Brief Communication

# Personal transcriptome variation is poorly explained by current genomic deep learning models

Connie Huang[1,4], Richard W. Shuai[1,4], Parth Baokar[1,4], Ryan Chung[2], Ruchir Rastogi[1], Pooja Kathail[2] & Nilah M. Ioannidis ●[1,2,3] ✉

Genomic deep learning models can predict genome-wide epigenetic features and gene expression levels directly from DNA sequence. While current models perform well at predicting gene expression levels across genes in different cell types from the reference genome, their ability to explain expression variation between individuals due to *cis*-regulatory genetic variants remains largely unexplored. Here, we evaluate four state-of-the-art models on paired personal genome and transcriptome data and find limited performance when explaining variation in expression across individuals. In addition, models often fail to predict the correct direction of effect of *cis*-regulatory genetic variation on expression.

With rapid advances in deep learning and growing datasets for training, there has been recent success in predicting gene expression levels[1–4], three-dimensional (3D) genome folding[5,6] and epigenetic features[7–10] such as transcription factor binding, histone modifications and chromatin accessibility directly from the reference genome sequence. These genomic deep learning models are trained using genome-wide data from a variety of cell types and cellular contexts and have been shown to learn biologically relevant regulatory motifs within the input DNA sequence[8,9]. Current sequence-to-expression models can explain variation in expression across different genes in the genome based on the reference genome sequence surrounding the start site of each gene. However, the application of such models to sequences from personal genomes to explain variation in gene expression across individuals (Fig. 1a) has been largely unexplored. Here, we evaluate four state-of-the-art models—Enformer (ref. 4), Basenji2 (ref. 11), ExPecto (ref. 2) and Xpresso (ref. 3)—on paired whole genome sequencing (WGS) and RNA sequencing (RNA-seq) data (*n* = 421) from the Geuvadis consortium[12] and show that model performance is limited when explaining gene expression variation across individuals. When the models do pick up on regulatory variation, for a limited set of genes, they often fail to capture the correct direction of effect of such variation on expression. Together with the recent findings of Sasse et al. [13], our work highlights shortcomings of

current deep learning models of gene expression when applied to personal genome interpretation.

To test these existing sequence-to-expression models on personal genome variation, we use RNA-seq data from the Geuvadis consortium, measured on lymphoblastoid cell lines (LCLs) and paired with WGS data from 421 individuals in the 1000 Genomes Project[14]. We focus on the 3,259 genes for which the Geuvadis analysis of expression quantitative loci (eQTLs) identified at least one statistically significant (FDR < 5%) genetic association where genotype of a *cis* variant is predictive of gene expression variation across individuals. We construct personal input sequences for each individual by inserting their single nucleotide variants (SNVs) into the reference sequence around each gene transcription start site (TSS). We then compute gene expression predictions for each individual, as well as for the reference genome sequence, using all four models (Methods). For each model, we use the output expression prediction track corresponding to the cell type most similar to the LCLs used for the Geuvadis measurements. To ensure that the chosen model outputs are indeed relevant for prediction of gene expression in LCLs, for each gene we compare the model prediction using the reference genome sequence with its median expression level in the Geuvadis dataset (Fig. 1b and Extended Data Fig. 1). We find Spearman rank correlations between reference predictions and observed expression levels of 0.57 for Enformer, 0.52 for Basenji2, 0.53 for ExPecto and 0.33 for

[1]Department of Electrical Engineering and Computer Sciences, University of California Berkeley, Berkeley, CA, USA. [2]Center for Computational Biology, University of California Berkeley, Berkeley, CA, USA. [3]Chan Zuckerberg Biohub, San Francisco, CA, USA. [4]These authors contributed equally: Connie Huang, Richard W. Shuai, Parth Baokar. ✉e-mail: nilah@berkeley.edu

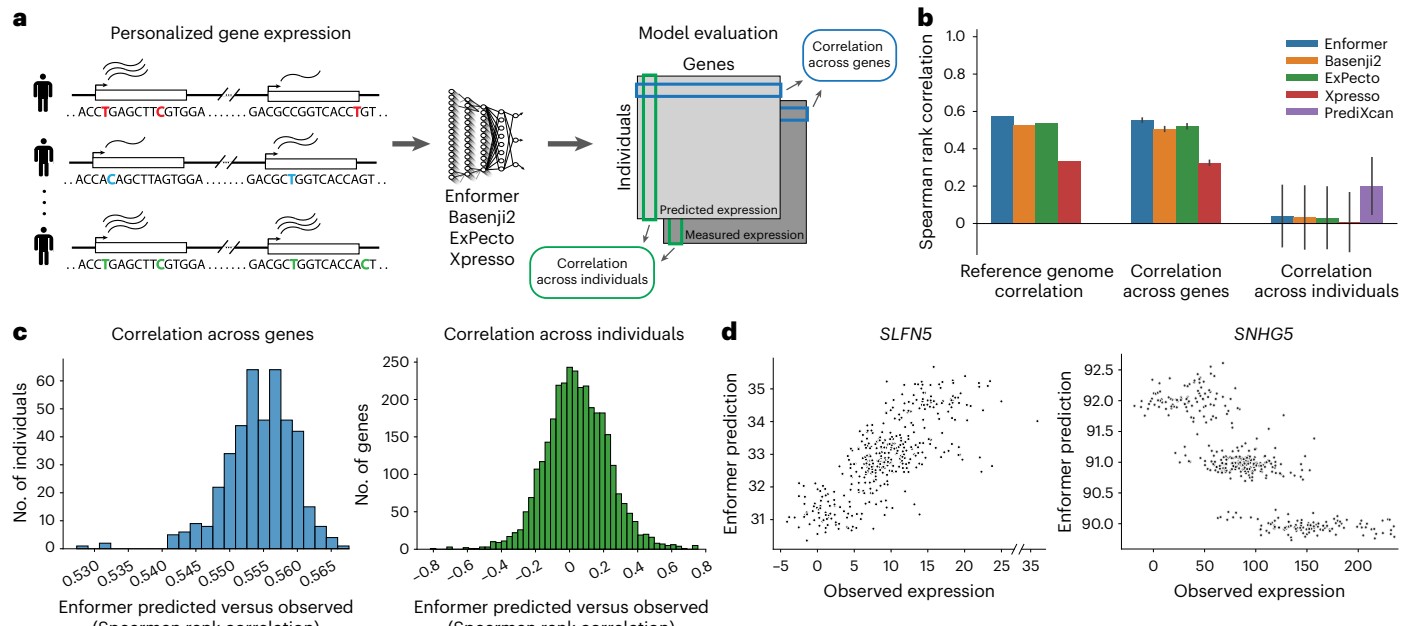

**Fig. 1 | Cross-gene versus cross-individual gene expression prediction.**
**a**, Overview of our approach, illustrating the cross-gene (blue) and cross-individual (green) measures of performance. Colored nucleotides on the left represent genetic variants present in each example individual. **b**, Performance of all tested models on reference sequence prediction, cross-gene prediction and cross-individual prediction. Bar heights represent means and error bars represent s.d. over all individuals ($n = 421$) for cross-gene Spearman rank correlation or over all genes ($n = 3,259$) for cross-individual Spearman rank

correlation. **c**, Distribution of Enformer cross-gene Spearman rank correlations for all individuals (left histogram) and Enformer cross-individual Spearman rank correlations for all genes (right histogram). Histograms for the other tested models are shown in Extended Data Figs. 2 and 3. **d**, Example genes with strong positive cross-individual correlation (*SLFN5*) and strong negative cross-individual correlation (*SNHG5*) of observed and predicted expression for Enformer.

Xpresso, indicating that these models explain a substantial fraction of expression variation across genes in LCLs, similar to previous reports.

For each model, we then compute two additional metrics using the personalized sequences as input. First, for each individual, we calculate a cross-gene correlation that compares the predicted expression levels of the aforementioned 3,259 genes using that individual's personal input sequence with the observed expression levels of those genes in the same individual. Similarly, for each gene, we compute a cross-individual correlation that compares the predicted expression levels in all 421 individuals with their observed expression levels (see Fig. 1a for a visual comparison of the two metrics). We find that the cross-gene correlation for each individual is similar to the reference genome performance of the corresponding model (Fig. 1b,c and Extended Data Fig. 2), with average Spearman correlations of 0.55 for Enformer, 0.51 for Basenji2, 0.52 for ExPecto and 0.32 for Xpresso. However, when we instead compute the correlation across individuals for each gene, we find that the distribution of cross-individual correlations is centered close to zero for all models (Fig. 1b,c, Extended Data Fig. 3 and Supplementary Fig. 1), indicating that all models struggle to explain variation in expression across individuals. This result suggests that current state-of-the-art sequence-to-expression models do not correctly predict the effects of many SNVs on gene expression. We also try ensembling the predictions from the four models and find that performance is improved only slightly by averaging predictions across models (Supplementary Fig. 2).

In comparison, regularized linear regression models trained separately for each gene using nearby variant dosages as predictors (the approach used by PrediXcan (ref. 15)) explain much more cross-individual variation, even when restricted to the same input context (197 kilobases (kb)) as Enformer (Fig. 1b and Extended Data Fig. 3). Since such PrediXcan-style models do not attempt to learn generalizable sequence features that can be applied to new sequences, variants or genes outside of the training set, we include these models

not as a competing approach, but rather as a minimum baseline for the genetic contribution to expression that should be possible to learn for each gene in the dataset. The higher performance of these PrediXcan-style models indicates effects of common *cis*-regulatory variants that are not captured by current deep learning models.

We also find that, although the mean cross-individual correlation is close to zero for all models, there are tails of strongly positively correlated and strongly negatively correlated genes for each model (Fig. 1c and Extended Data Fig. 3). Example genes with strong positive correlation and strong negative correlation are shown for Enformer in Fig. 1d. When comparing predictions for such genes across all four models, we find that the models often disagree with one another on the direction of correlation (Fig. 2a,b, Extended Data Fig. 4 and Supplementary Fig. 3). This result suggests that the incorrectly predicted direction of genetic effect for the negatively correlated genes for any given model is not due to an inherent difficulty in modeling those particular genes or their corresponding variants, but rather to noise in the effects attributed to variants by these types of models. Importantly, we find that the four tested models are more consistent with one another in the magnitude of their correlation to observed expression of a given gene than in the direction of that correlation (Fig. 2b and Extended Data Fig. 4), suggesting that they agree on identifying causal regulatory variants more than they agree on the direction of effect of such variants on expression.

We next explore whether predicted directions of genetic effect on expression tend to be more accurate for certain types of genes. First, we test whether genes with strong genetic associations in the Geuvadis eQTL analysis are more likely to have correctly predicted directions of genetic effect by comparing the cross-individual correlation for each gene with the *P* value (Fig. 2c and Extended Data Fig. 5), effect size (Extended Data Fig. 6) and minor allele frequency (MAF) (Extended Data Fig. 7) of the most statistically significant eQTL within 20 kb of the TSS. We find that genes with strong eQTLs tend to have larger magnitude cross-individual correlations for all

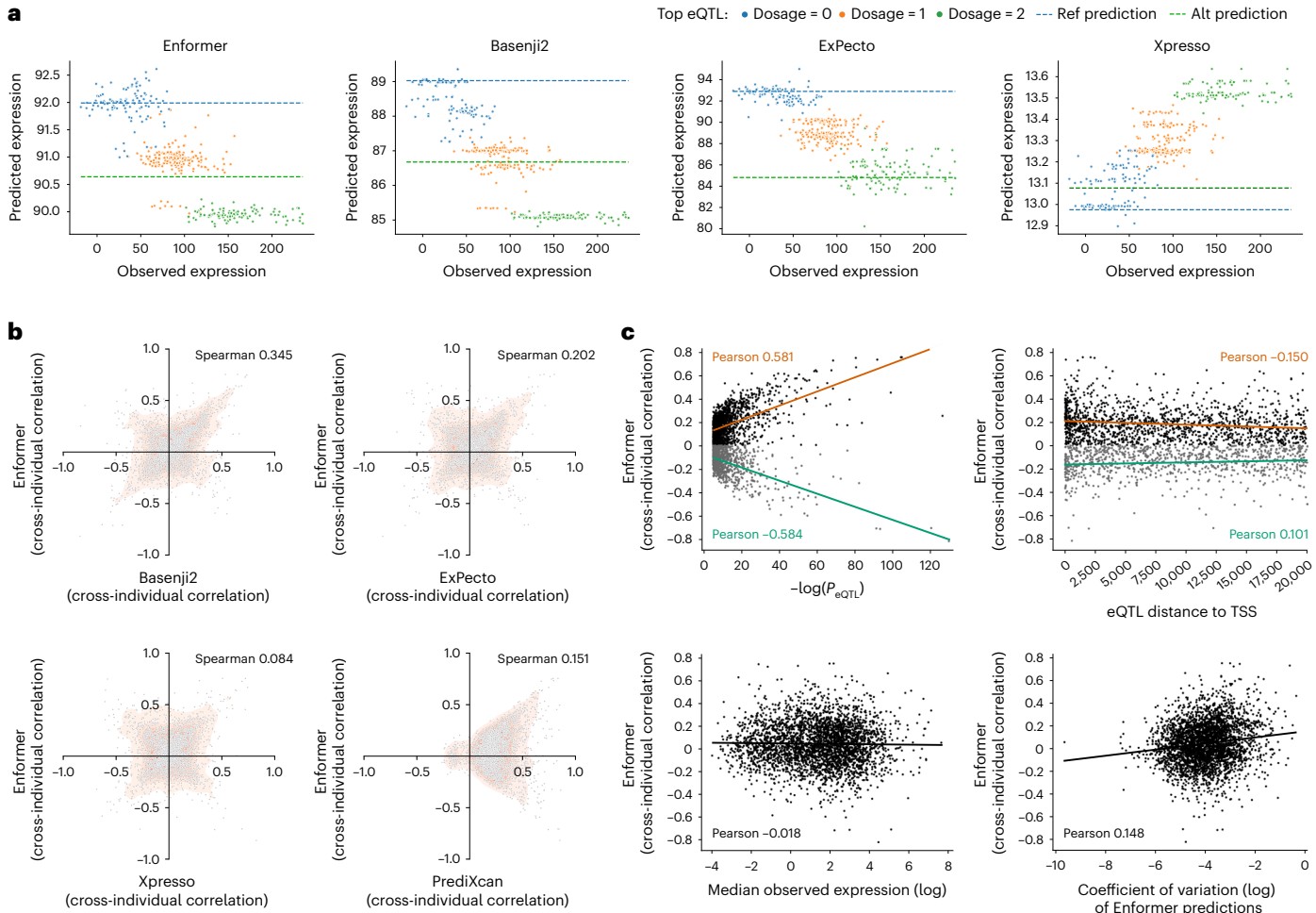

Fig. 2 | **Models often disagree on predicted direction of effect of *cis*-regulatory variation. a**, Predictions from all four deep learning models on an example gene, *SNHG5*, that has strong negative cross-individual correlations for Enformer, Basenji2 and ExPecto, and positive cross-individual correlation for Xpresso. Points are colored by the corresponding individual's dosage of the most statistically significant eQTL for this gene. Dashed lines indicate the predicted expression levels of the reference (Ref) and alternate (Alt) alleles of the most statistically significant eQTL. **b**, Comparison of cross-individual Spearman rank correlations for Enformer versus other models. A kernel density estimate of each scatterplot is overlaid (red). Note the increased density of genes along the $y = x$ and $y = -x$ axes. Related plots for all pairs of models are shown in Extended Data

Fig. 4. **c**, Cross-individual Spearman rank correlations for Enformer compared with the *P* value of the most statistically significant eQTL in each gene (top left), the distance to the TSS for that eQTL (top right), the median observed expression level of the gene (bottom left) and the coefficient of variation of the predicted expression levels of the gene (bottom right). Note that negative cross-individual correlations are observed even for genes with strong eQTLs. For each plot, Pearson correlations and lines of best fit using ordinary least squares are shown in black when computed using all genes, and in orange or green when computed using only genes with positive or negative cross-individual correlations, respectively. Related plots for all tested models are shown in Extended Data Figs. 5–10.

models; however, these genes are not more likely to have positive rather than negative cross-individual correlations, indicating that the models often predict incorrect directions of effect even for genes with strong genetic effects on expression. We find a small trend towards larger cross-individual correlations for genes with smaller distance between the most statistically significant eQTL and the TSS (Fig. 2c and Extended Data Fig. 8), which aligns with previous findings that current sequence-to-expression models capture gene expression determinants in promoters more accurately than distal enhancers[16]. However, we note that genes with proximal eQTLs still frequently have strong negative cross-individual correlations, suggesting that modeling distal regulatory effects and predicting regulatory effect direction are two important, but orthogonal, areas for future modeling improvements. Last, we find only small trends when comparing model performance with the median observed expression level of a gene (Fig. 2c and Extended Data Fig. 9) and with the variation in predicted expression levels across individuals (Fig. 2c and Extended Data Fig. 10).

In conclusion, we analyze the performance of four state-of-the-art sequence-to-expression deep learning models—Enformer, Basenji2, ExPecto and Xpresso—on personalized gene expression prediction, and find that these models consistently under-perform when predicting differences in expression for a given gene across individuals based on inter-individual variation in the input DNA sequence. We also find genes with strong negative correlations between predicted and observed expression levels, for which the models have probably identified causal regulatory variant(s) but incorrectly predicted their direction of effect. Previous evaluations of variant effect prediction with sequence-to-expression deep learning models have focused on individual variant effects, as measured by eQTL studies, or massively parallel reporter assays. However, massively parallel reporter assays lack the complex genomic and chromatin context of endogenous gene expression, and it is difficult to identify the causal variants in eQTL studies, even with current fine-mapping approaches, resulting in effect size estimates that are not biologically meaningful for variants

that are in linkage disequilibrium with a causal variant. By using personal genome sequences to evaluate model performance, our input sequences include all variants surrounding the TSS for each individual and thus avoid the issue of causal variant identification.

Our conclusions about directionality prediction are in line with previous tests on eQTLs[2,4], which showed low performance on predicting the direction of effect on expression for individual variants, especially for distal eQTLs. Following Avsec et al.[4], we confirm this finding for Enformer for fine-mapped GTEx eQTLs in LCLs (Supplementary Fig. 4). Our preliminary analysis also suggests that these models have room for improvement in predicting the direction of effect of chromatin accessibility quantitative trait loci (caQTLs) as well (Supplementary Fig. 5), although further work is needed to evaluate the ability of genomic deep learning models to explain cross-individual variation in accessibility and other molecular phenotypes, as discussed below.

Finally, our cross-model analysis reveals that models often strongly disagree with one another on the predicted direction of genetic effects on expression and, intriguingly, that agreement between models is greater for the magnitude of cross-individual correlation than the direction of that correlation. This result further supports the conclusion that current genomic deep learning models recognize the presence of important regulatory variation in an input sequence but struggle with understanding the direction of effect of such variation. To diagnose the reasons for these errors, it will be valuable to assess whether model predictions of variant effects on other epigenetic tracks (for example, transcription factor binding and chromatin accessibility) are more accurate than for gene expression. For example, these models may have correctly learned variant effects on more proximal phenotypes, such as individual regulatory elements, but struggle to map effects of those elements to corresponding changes in gene expression; alternatively, the models may struggle with direction of variant effects even on proximal phenotypes such as the binding of individual transcription factors. Further work to distinguish between these possibilities will help prioritize future modeling improvements to focus on understanding high-level regulatory grammar (for example, through hierarchical models of gene expression), or to focus on more accurately learning local variant effects (for example, by increasing sequence diversity during model training).

## Online content

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

## Methods

All data used in this study are publicly available ('Data availability'), and no specific ethical approval was required.

### Gene expression dataset

The data used to evaluate gene expression predictions for personal genome sequences were obtained from the Geuvadis consortium[12], which includes paired gene expression and WGS data from individuals in the 1000 Genomes Project[14]. The E-GEUV-1 release includes RNA-seq data from LCLs from a total of 465 samples. After excluding samples with unphased imputed genotypes, there were 421 Geuvadis individuals with phased WGS data that we used for our analysis. These samples originated from five populations with ancestry in Europe and Africa: 92 Tuscan (TSI), 89 Finnish (FIN), 85 British (GBR), 78 European from Utah (CEU) and 77 Yoruban (YRI). We also obtained results from the Geuvadis *cis*-eQTL analysis performed in European individuals ($n = 373$), which included autosomal protein-coding and lincRNA genes from GENCODE v.12 and considered variants with MAF > 5% located within 1 Mb of a gene TSS. Except where otherwise noted, our results are shown for all 3,259 genes that had a statistically significant eQTL (false discovery rate < 5%) in the Geuvadis EUR *cis*-eQTL analysis.

### Comparison of deep learning models for gene expression prediction

We test four state-of-the-art deep learning models that make gene expression predictions for an input DNA sequence. These models consider different sequence contexts, or receptive fields, when making predictions; in particular, Enformer has the widest receptive field (98.3 kb upstream and 98.3 kb downstream of the gene TSS), followed by Basenji2 (27.5 kb upstream and 27.5 kb downstream), ExPecto (20 kb upstream and 20 kb downstream) and Xpresso (7 kb upstream and 3.5 kb downstream). All models include standard convolutional layers, with additional dilated convolutional layers in Basenji2 and transformer layers in Enformer. The models also use different sources of gene expression data during training; in particular, Basenji2 and Enformer are trained using genome-wide cap analysis of gene expression (CAGE) measurements, while ExPecto and Xpresso are trained using RNA-seq data. Basenji2 and Enformer use multitask learning to make gene expression predictions along with many other epigenetic track predictions in a variety of cell types, while Xpresso predicts gene expression alone. ExPecto uses a hierarchical model, making predictions of epigenetic tracks along the input sequence and then adding a linear transformation on top of those outputs to predict expression.

### Constructing personalized input sequences

For each of the 3,259 genes from the Geuvadis analysis mentioned above, the ENSEMBL gene ID, TSS position from GENCODE v.12, strand and chromosome were obtained from Geuvadis. We used hg19 as the reference genome for creating personalized sequences, to match the Geuvadis dataset. For ExPecto, Basenji2 and Enformer, whose receptive fields are symmetric about the TSS, and for genes located on the positive strand for Xpresso, we computed personalized sequences directly around the TSS using bcftools consensus[17]. Since Xpresso uses an asymmetric input sequence, for genes located on the negative strand, we extracted the reference sequence 3.5 kb before the TSS to 7 kb after the TSS using Samtools[17], applied bcftools consensus and then took the reverse complement. We considered only SNVs and did not include indels when creating the personalized input sequences. We predicted gene expression levels as described below for the two personalized haplotypes for each individual and averaged the predictions from both haplotypes.

### Basenji2 predictions

Basenji2 takes input sequences of 131 kb with an effective receptive field of 55 kb for each prediction. The model outputs predictions in 128-bp

bins for 5,313 epigenetic and transcriptional tracks from the ENCODE, Roadmap Epigenomics and FANTOM consortiums. We used Basenji2 predictions for the GM12878 LCL CAGE track, as it is the cell type most relevant for the Geuvadis expression data. For a given input sequence centered at a gene TSS, we averaged predictions from the forward and reverse complement sequence as well as minor sequence shifts to the left and right (one nucleotide in each direction). To compute the final expression prediction for a gene, we averaged the predicted CAGE signal over a ten-bin window around each TSS.

### Enformer predictions

Enformer replaces the dilated convolutions of Basenji2 with a self-attention mechanism, which facilitates the learning of long-range dependencies. Enformer has a receptive field of 196.6 kb and outputs predictions in 128-bp bins for the same 5,313 tracks as Basenji2. As above, we used Enformer predictions for CAGE measurements performed on GM12878. While the Enformer authors averaged predictions within a three-bin window around each gene TSS, we found that averaging over a ten-bin window led to better performance on the Geuvadis dataset.

### ExPecto predictions

ExPecto predicts gene expression by first using a convolutional neural network (Beluga, an updated version of DeepSEA (ref. [7])) to predict chromatin features within a 40 kb region around each gene TSS. Specifically, Beluga outputs predictions in 200-bp bins for 2,002 epigenetic tracks from the ENCODE and Roadmap consortiums. To predict expression for a given gene, Beluga is used to predict chromatin features for 200 bins centered around the TSS, averaging predictions over the input sequence and its reverse complement. The resulting predictions are spatially transformed with a set of basis functions and used as input features for an L2-regularized linear regression model to predict expression for the given input sequence. For our ExPecto predictions, we used a publicly available ExPecto model trained on Epstein-Barr virus (EBV)-transformed lymphocytes from GTEx, which we chose as the most relevant cell type to compare with the Geuvadis expression data.

### Xpresso predictions

Xpresso consists of two convolutional blocks and two fully connected layers trained on normalized RNA-seq data across 56 tissues and cell lines from the Roadmap Epigenomics Consortium. The optimal input sequence for Xpresso was found to be a 10.5 kb region asymmetrically centered around the TSS. For our analysis, we used predictions from the pretrained LCL-specific Xpresso model as the most relevant to the Geuvadis dataset.

### Elastic net (PrediXcan-style) gene expression model

For comparison with the sequence-to-expression deep learning models above, we also trained a cross-validated, elastic net regression model for each gene to predict Geuvadis expression measurements from common variants (MAF ≥ 0.05) within 98.3 kb of the TSS to match Enformer's receptive field. We set the elastic net mixing parameter to 0.5 and found the best regularization penalty in a tenfold cross-validation scheme using scikit-learn's ElasticNetCV (ref. [18]) in Python. We obtained individual gene expression predictions from the output of ten holdout validation splits, such that the model makes predictions on individuals not in the training set.

### QTL effect direction classification

We obtained GTEx v.8 eQTLs fine-mapped using the SuSiE method[19,20] from the Supplementary Data in Avsec et al.[4]. Using these data, we evaluated Enformer on its ability to predict the direction of eQTL effect on expression. We used fine-mapped eQTLs identified in EBV-transformed lymphocytes as the GTEx cell type that matched the

Geuvadis expression data most closely. Using a method similar to that described in Avsec et al.[4], we focused on variants with a posterior inclusion probability (PIP) in a credible causal set of greater than 0.9, and removed variants that affect gene expression in opposite directions for different *cis* genes. We used Enformer predictions from the GM12878 CAGE track, as the closest match to EBV-transformed lymphocytes. We computed expression direction accuracy over 100 bootstrap samples from the full set of variants. We also obtained caQTL variants from the Tehranchi et al.[21] analysis of ATAC-seq data from LCLs from individuals in ten populations: four African, four European, one African-American and one Han Chinese. Using these data, we evaluated Enformer on its ability to predict the direction of caQTL effect on accessibility. To generate accessibility predictions, we summed Enformer's predictions from the three 128-bp bins closest to the variant for the DNase:GM12878 track (track number 69). For computational efficiency, we randomly sampled 10,000 caQTLs from the Tehranchi et al.[21] dataset, and then computed effect direction accuracy over 100 bootstrap samples of those 10,000 variants.

### Statistics and reproducibility
No statistical method was used to predetermine sample size. All individuals with RNA-seq and phased WGS data from the Geuvadis consortium were included. All genes that were found to have at least one statistically significant (FDR < 5%) *cis*-eQTL association in the Geuvadis analysis were included. All code to reproduce our analyses is publicly available ('Code availability').

### Reporting summary
Further information on research design is available in the Nature Portfolio Reporting Summary linked to this article.

## Data availability
The Geuvadis gene expression data and WGS data used in this study are publicly available at https://www.ebi.ac.uk/biostudies/arrayexpress/studies/E-GEUV-1.

## Code availability
All model predictions on Geuvadis individuals and scripts to generate personalized sequences, get model predictions and plot figures are available at https://github.com/ni-lab/personalized-expression-benchmark (ref. 22).

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

## Acknowledgements
We thank G. Loeb, D. Kelley and members of the Ioannidis laboratory for helpful discussions. This work was partially supported by the US National Institutes of Health grant R00HG009677 to N.M.I., an Okawa Foundation Research Grant to N.M.I. and a grant from the UC Noyce Initiative for Computational Transformation to N.M.I. N.M.I. is a Chan Zuckerberg Biohub San Francisco Investigator. The funders had no role in study design, data collection and analysis, decision to publish or preparation of the manuscript.

## Author contributions
C.H., R.W.S., P.B., R.C., R.R. and P.K. performed the analyses. N.M.I. conceived and supervised the study. All authors interpreted the results. All authors wrote and edited the manuscript.

## Competing interests
The authors declare no competing interests.

## Additional information
**Extended data** is available for this paper at https://doi.org/10.1038/s41588-023-01574-w.

**Correspondence and requests for materials** should be addressed to Nilah M. Ioannidis.

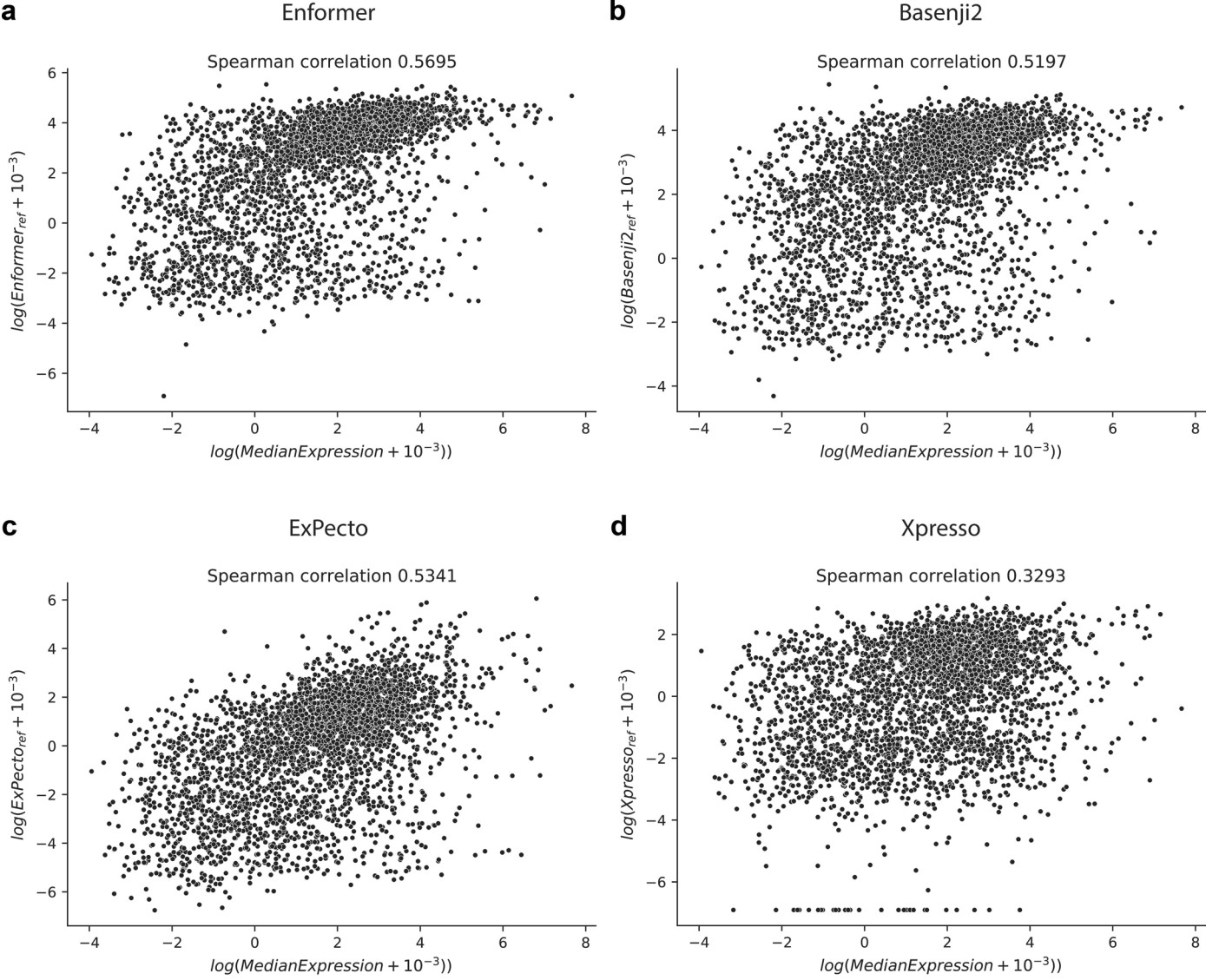

**Extended Data Fig. 1 | Performance of all tested models on reference sequence prediction.** Median Geuvadis gene expression (log transformed) versus gene expression predictions (log transformed) obtained by inputting the reference genome sequence to **(a)** Enformer, **(b)** Basenji2, **(c)** ExPecto, and **(d)** Xpresso. For each model, gene expression predictions from the most relevant cell type were used, as described in Methods. Measurements and predictions for the 3,259 genes with at least one statistically signficant (FDR < 5%) eQTL in the Geuvadis analysis are displayed.

**a**

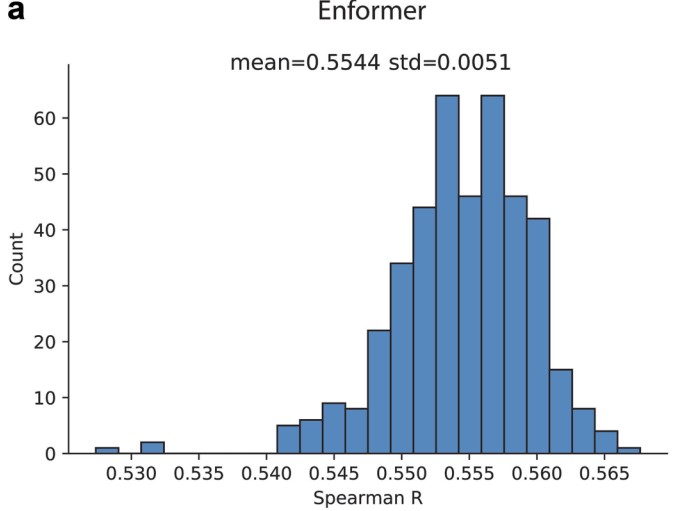

**b**

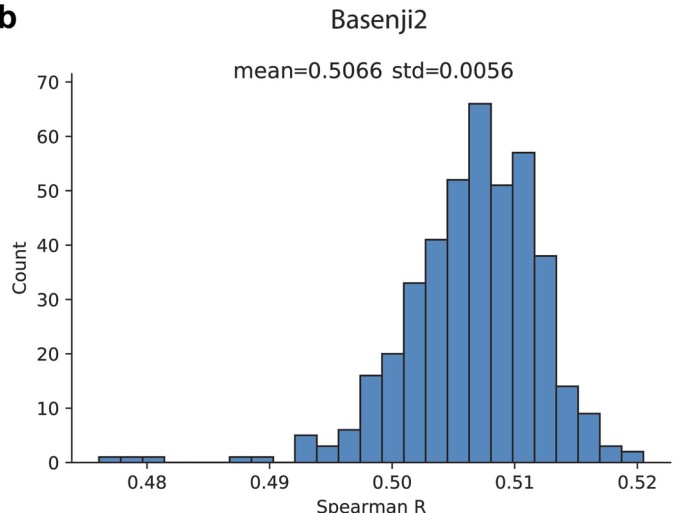

**c**

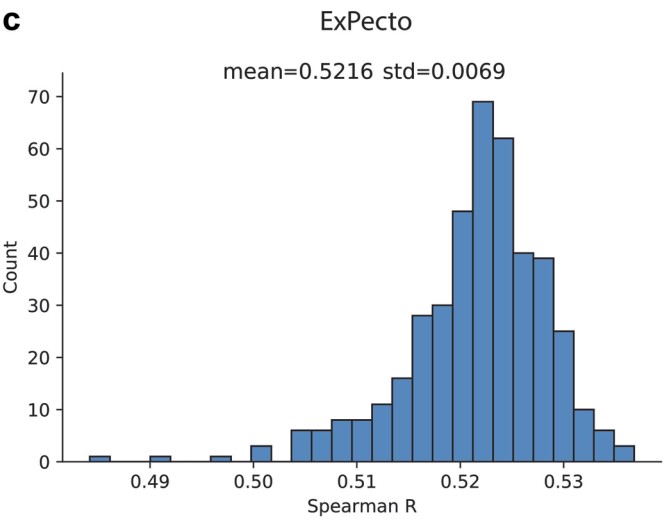

**d**

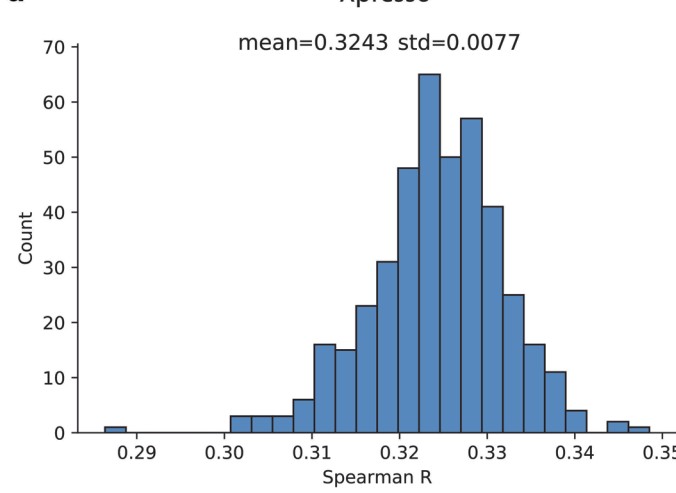

**Extended Data Fig. 2 | Performance of all tested models on cross-gene prediction.** Cross-gene performance for **(a)** Enformer, **(b)** Basenji2, **(c)** ExPecto, and **(d)** Xpresso. For a given individual, cross-gene performance is defined as the correlation between their measured gene expression levels and gene expression predictions obtained using their personalized genome sequences. Correlations were computed across the 3,259 genes with at least one statistically signficant (FDR < 5%) eQTL in the Geuvadis analysis. Each histogram displays the distribution of cross-gene performance over all individuals.

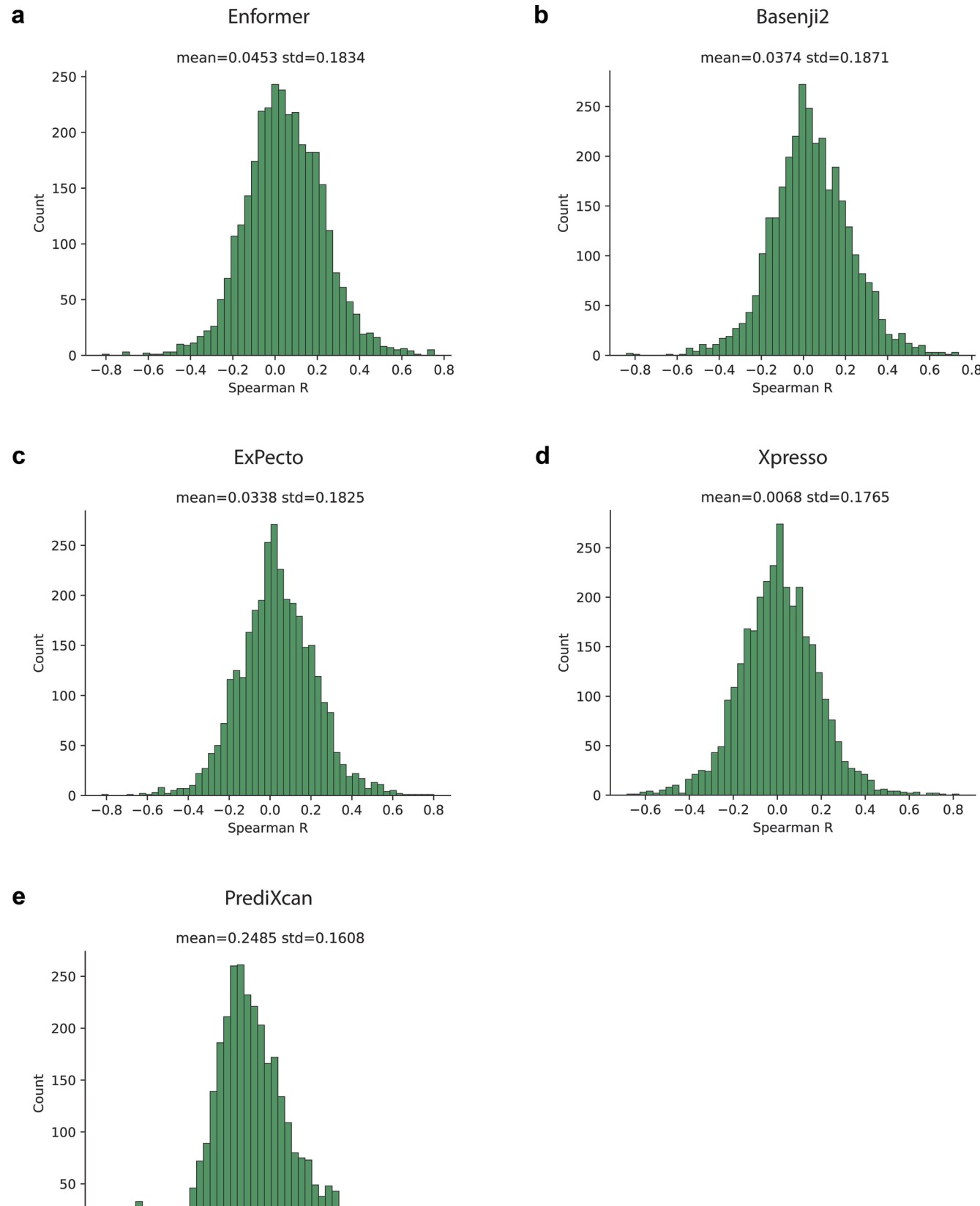

**Extended Data Fig. 3 | Performance of all tested models on cross-individual prediction.** Cross-individual performance for **(a)** Enformer, **(b)** Basenji2, **(c)** ExPecto, **(d)** Xpresso, and **(e)** PrediXcan. For a given gene, cross-individual performance is defined as the correlation between measured gene expression levels in all 421 individuals and corresponding gene expression predictions obtained using each individual's personalized genome sequence. Each histogram displays the distribution of cross-individual performance for the 3,259 genes with at least one statistically signficant (FDR < 5%) eQTL in the Geuvadis analysis.

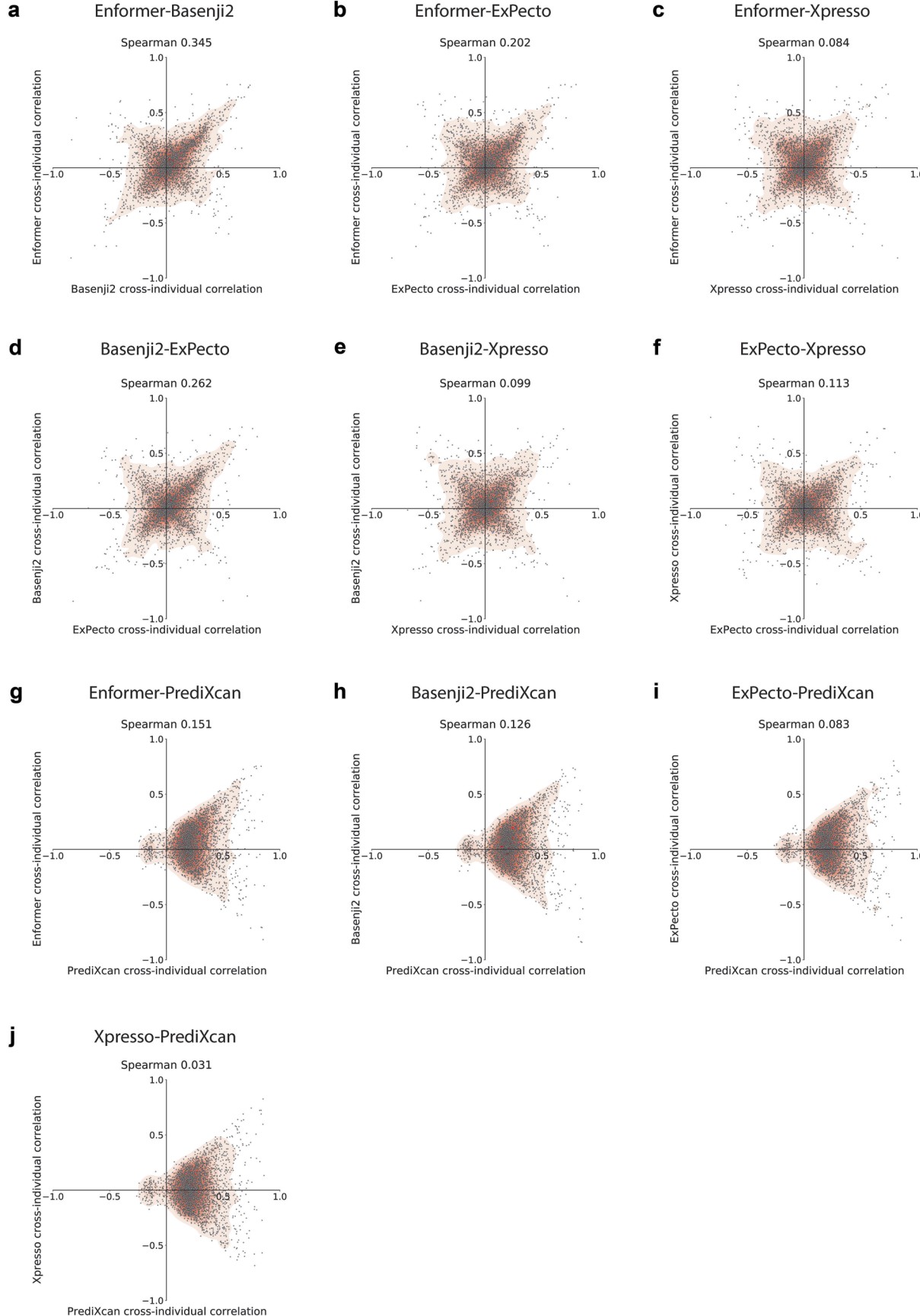

**Extended Data Fig. 4 | See next page for caption.**

**Extended Data Fig. 4 | Pairwise model comparisons of cross-individual correlation.** Comparison of cross-individual Spearman correlations between each pair of models: (a) Enformer & Basenji2, (b) Enformer & ExPecto, (c) Enformer & Xpresso, (d) Basenji2 & ExPecto, (e) Basenji2 & Xpresso, (f) ExPecto & Xpresso, (g) Enformer & PrediXcan, (h) Basenji2 & PrediXcan, (i) ExPecto & PrediXcan, and (j) Xpresso & PrediXcan. The scatterplots display, for each gene, the performance achieved by both models. A kernel density estimate of each scatterplot is overlaid (red). Note the increased density of genes along the y = x and y = -x axes.

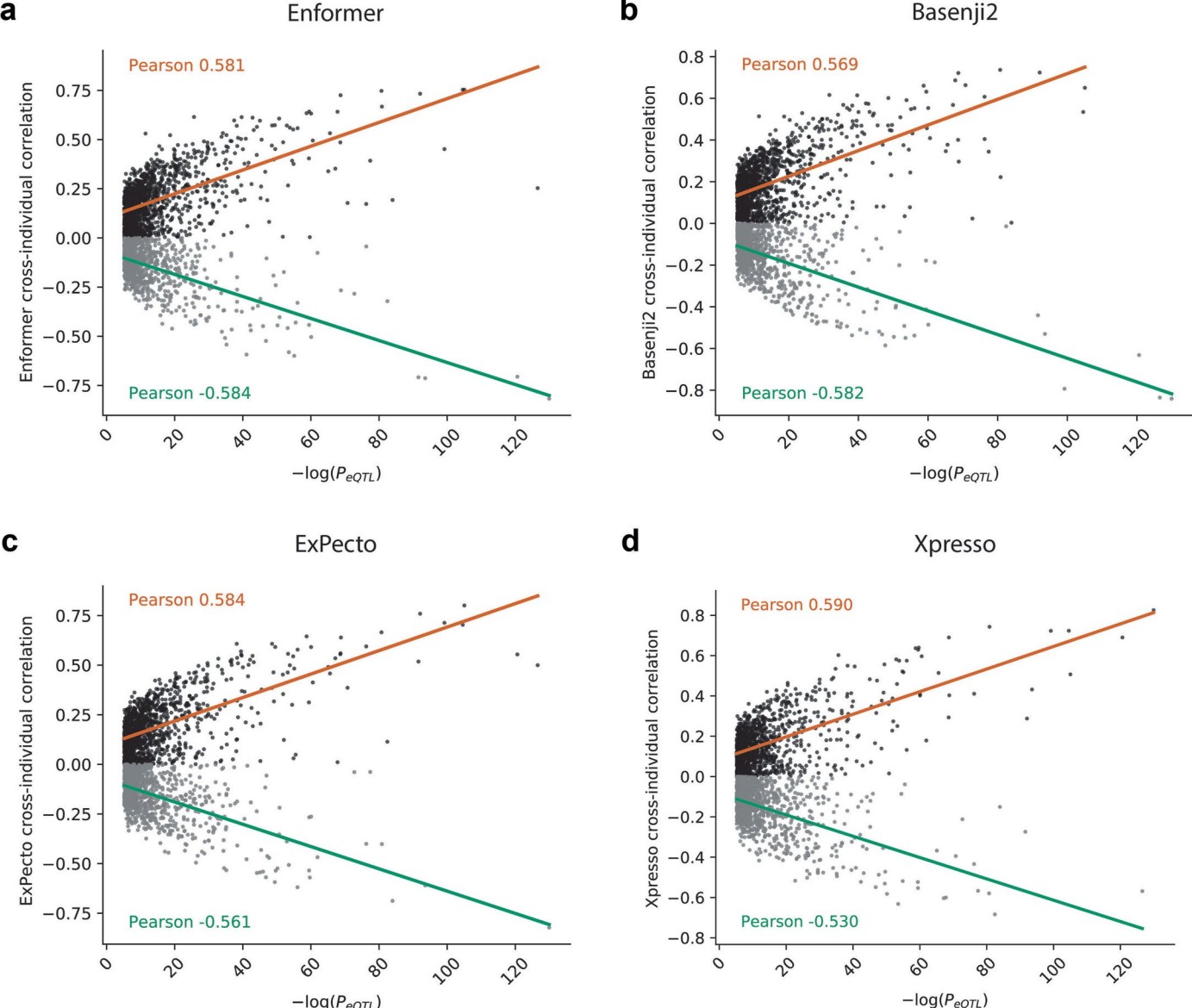

**Extended Data Fig. 5 | Cross-individual correlation vs. top eQTL p-value for all tested models.** Cross-individual correlations for **(a)** Enformer, **(b)** Basenji2, **(c)** ExPecto, and **(d)** Xpresso compared to the p-value of the most statistically significant Geuvadis eQTL in each gene. For each model, the Pearson correlation and line of best fit using ordinary least squares are shown separately for genes with positive and negative cross-individual correlations (orange and green, respectively).

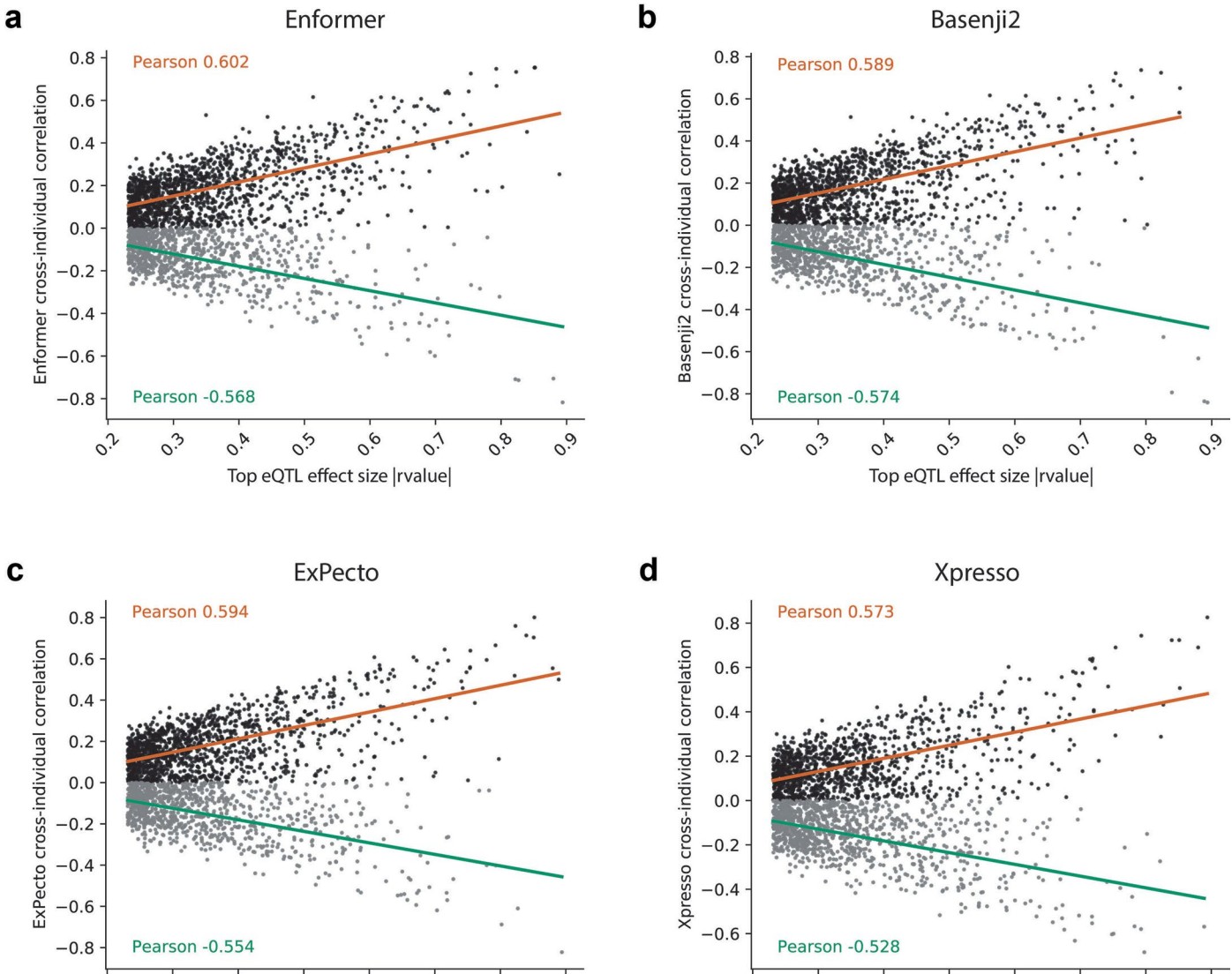

**Extended Data Fig. 6 | Cross-individual correlation vs. top eQTL effect size for all tested models.** Cross-individual correlations for **(a)** Enformer, **(b)** Basenji2, **(c)** ExPecto, and **(d)** Xpresso compared to the absolute value of the effect size of the most statistically significant Geuvadis eQTL in each gene. For each model, the Pearson correlation and line of best fit using ordinary least squares are shown separately for genes with positive and negative cross-individual correlations (orange and green, respectively).

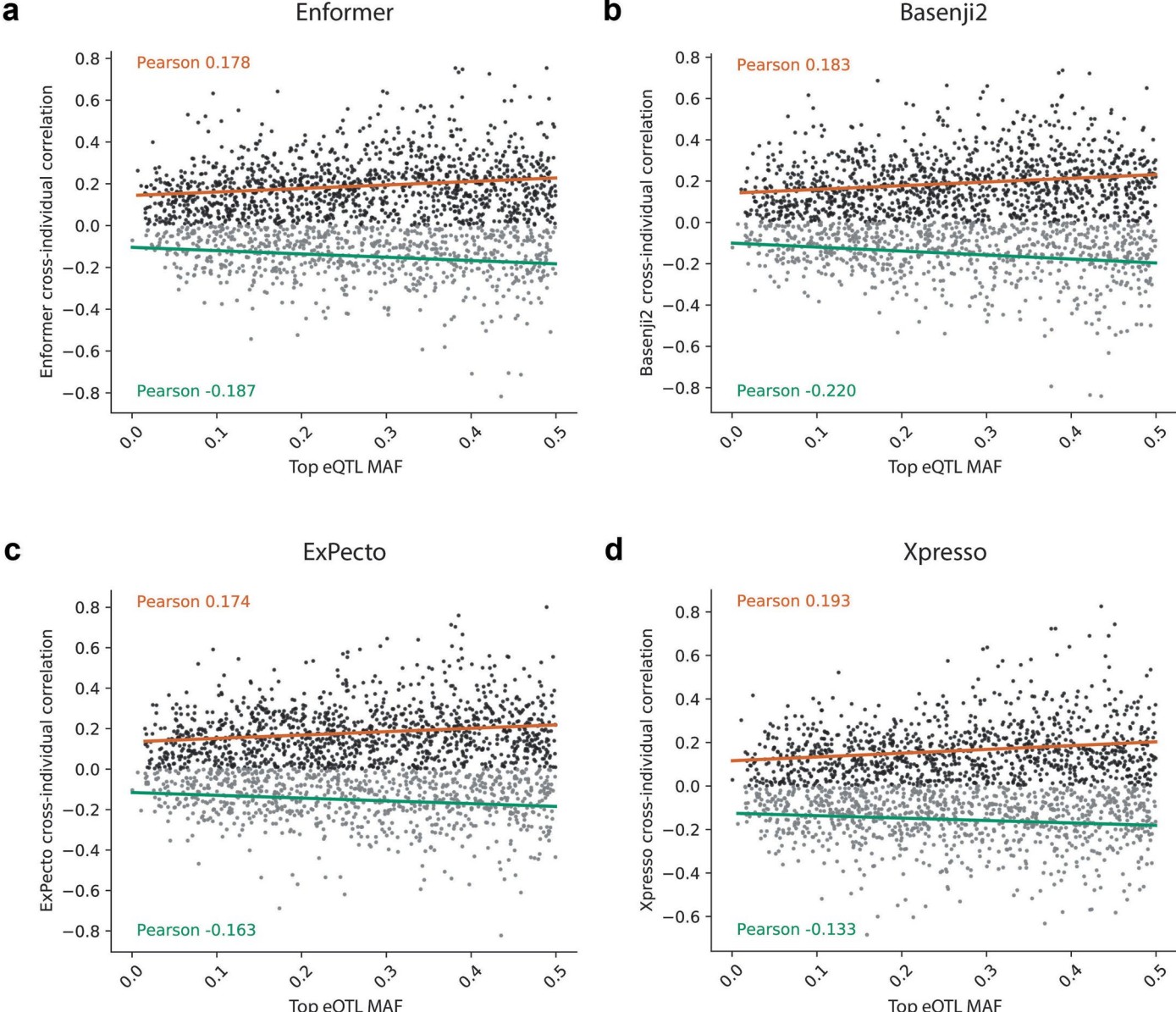

**Extended Data Fig. 7 | Cross-individual correlation vs. top eQTL allele frequency for all tested models.** Cross-individual correlations for **(a)** Enformer, **(b)** Basenji2, **(c)** ExPecto, and **(d)** Xpresso compared to the global minor allele frequency (from Ensembl biomaRt) of the most statistically significant Geuvadis eQTL in each gene. For each model, the Pearson correlation and line of best fit using ordinary least squares are shown separately for genes with positive and negative cross-individual correlations (orange and green, respectively).

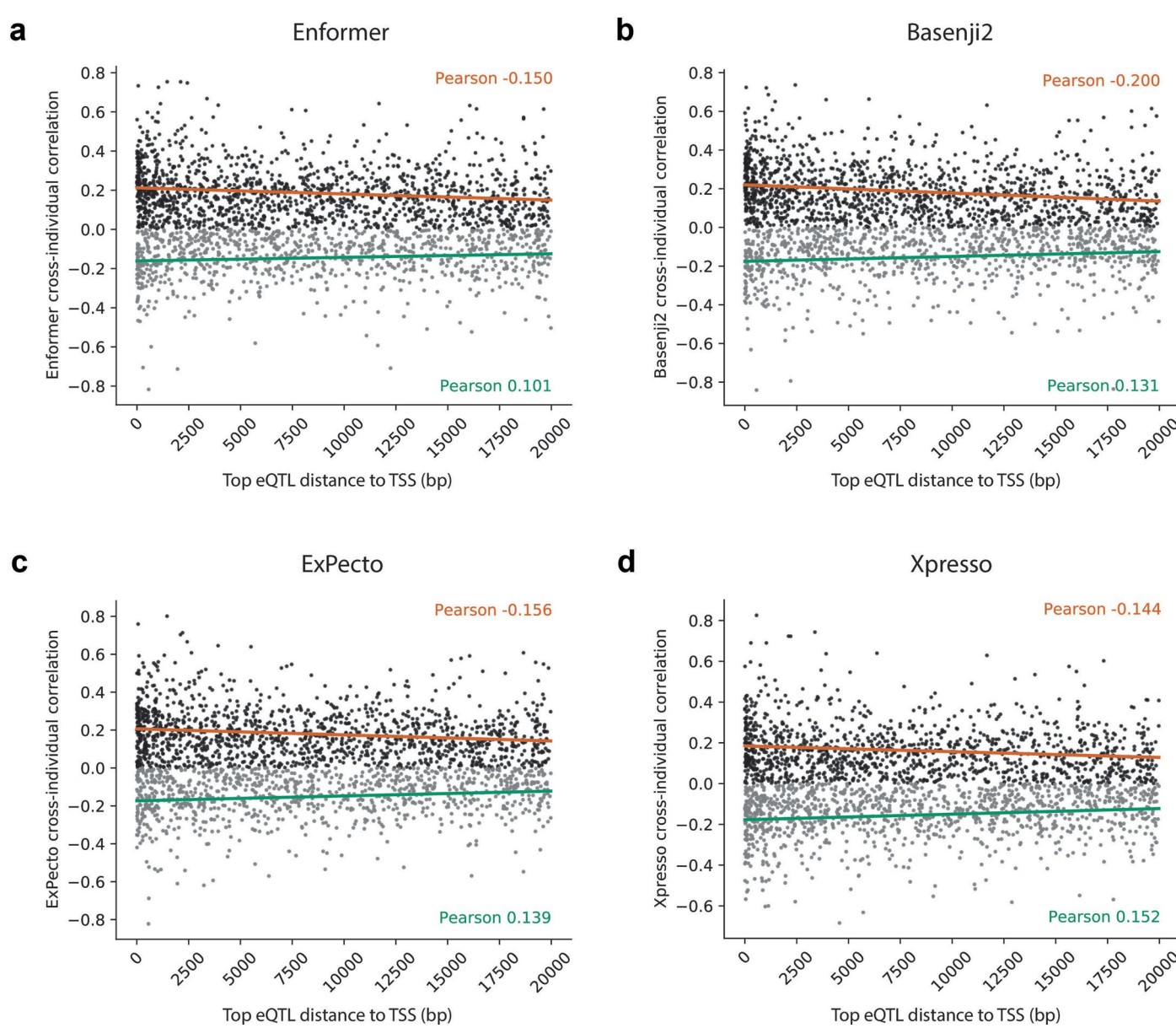

**Extended Data Fig. 8 | Cross-individual correlation vs. top eQTL distance to TSS for all tested models.** Cross-individual correlations for **(a)** Enformer, **(b)** Basenji2, **(c)** ExPecto, and **(d)** Xpresso compared to the distance between each gene's TSS and its most statistically significant Geuvadis eQTL. For each model, the Pearson correlation and line of best fit using ordinary least squares are shown separately for genes with positive and negative cross-individual correlations (orange and green, respectively).

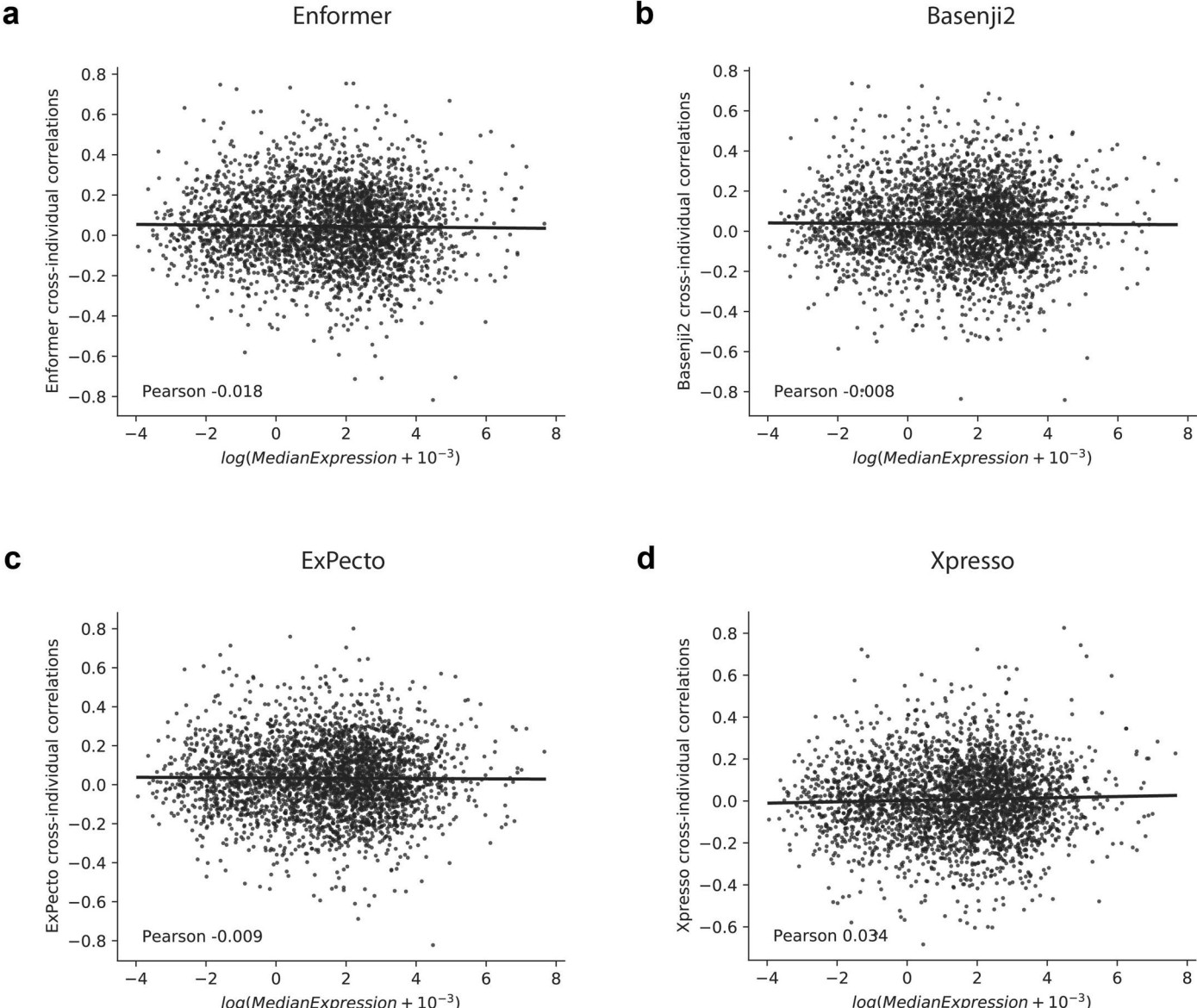

**Extended Data Fig. 9 | Cross-individual correlation vs. median gene expression for all tested models.** Cross-individual correlations for **(a)** Enformer, **(b)** Basenji2, **(c)** ExPecto, and **(d)** Xpresso compared to the median Geuvadis gene expression level (log transformed) for each gene. For each model, the Pearson correlation and line of best fit using ordinary least squares are shown.

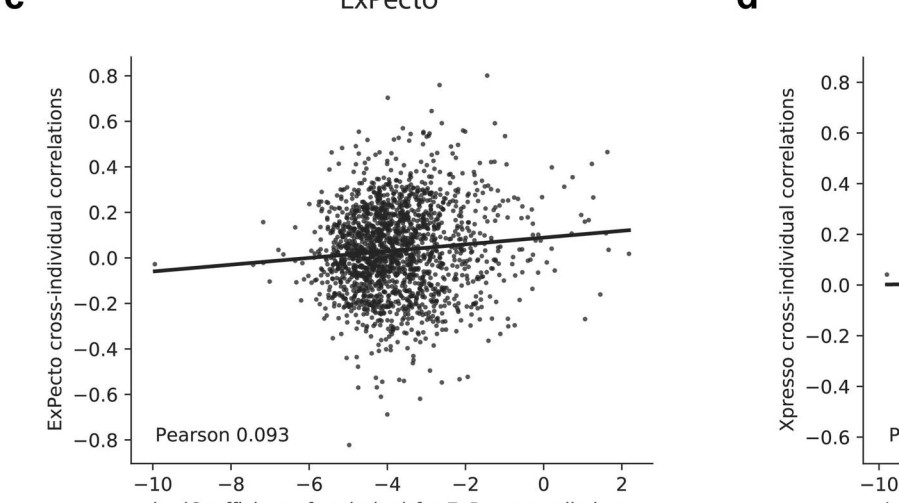
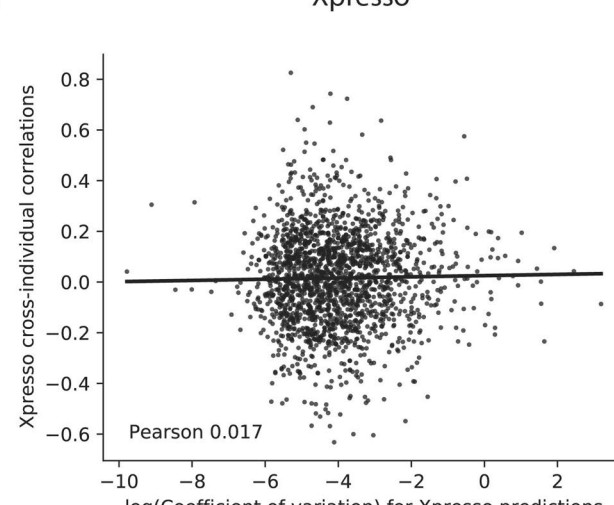

**Extended Data Fig. 10 | Cross-individual correlation vs. predicted expression dispersion for all tested models.** Cross-individual correlations for **(a)** Enformer, **(b)** Basenji2, **(c)** ExPecto, and **(d)** Xpresso compared to the log coefficient of variation ($\log \sigma/\mu$), a measure of dispersion, in the model predictions for each gene. For each model, the Pearson correlation and line of best fit using ordinary least squares are shown.

nature portfolio

# Reporting Summary

## Statistics

For all statistical analyses, confirm that the following items are present in the figure legend, table legend, main text, or Methods section.

| n/a | Confirmed | |
|---|---|---|
| ☐ | ☒ | The exact sample size (*n*) for each experimental group/condition, given as a discrete number and unit of measurement |
| ☐ | ☒ | A statement on whether measurements were taken from distinct samples or whether the same sample was measured repeatedly |
| ☐ | ☒ | The statistical test(s) used AND whether they are one- or two-sided *Only common tests should be described solely by name; describe more complex techniques in the Methods section.* |
| ☒ | ☐ | A description of all covariates tested |
| ☒ | ☐ | A description of any assumptions or corrections, such as tests of normality and adjustment for multiple comparisons |
| ☐ | ☒ | A full description of the statistical parameters including central tendency (e.g. means) or other basic estimates (e.g. regression coefficient) AND variation (e.g. standard deviation) or associated estimates of uncertainty (e.g. confidence intervals) |
| ☐ | ☒ | For null hypothesis testing, the test statistic (e.g. *F*, *t*, *r*) with confidence intervals, effect sizes, degrees of freedom and *P* value noted *Give P values as exact values whenever suitable.* |
| ☒ | ☐ | For Bayesian analysis, information on the choice of priors and Markov chain Monte Carlo settings |
| ☒ | ☐ | For hierarchical and complex designs, identification of the appropriate level for tests and full reporting of outcomes |
| ☐ | ☒ | Estimates of effect sizes (e.g. Cohen's *d*, Pearson's *r*), indicating how they were calculated |

*Our web collection on statistics for biologists contains articles on many of the points above.*

## Software and code

Policy information about availability of computer code

| Data collection | No software was used for data collection. |
|---|---|
| Data analysis | We used publicly available code to run Enformer, Basenji2, ExPecto, and Xpresso. Custom code for our analysis is available at https://github.com/ni-lab/personalized-expression-benchmark. We also used Python version 3.6.13, scikit-learn 0.24.2, bcftools 1.14, and samtools 1.14. |

For manuscripts utilizing custom algorithms or software that are central to the research but not yet described in published literature, software must be made available to editors and reviewers. We strongly encourage code deposition in a community repository (e.g. GitHub). See the Nature Portfolio guidelines for submitting code & software for further information.

## Data

Policy information about availability of data

All manuscripts must include a data availability statement. This statement should provide the following information, where applicable:

- Accession codes, unique identifiers, or web links for publicly available datasets
- A description of any restrictions on data availability
- For clinical datasets or third party data, please ensure that the statement adheres to our policy

The gene expression and whole genome sequencing data used in this study are publicly available at https://www.ebi.ac.uk/biostudies/arrayexpress/studies/E-GEUV-1.

# Research involving human participants, their data, or biological material

Policy information about studies with human participants or human data. See also policy information about sex, gender (identity/presentation), and sexual orientation and race, ethnicity and racism.

| | |
|---|---|
| Reporting on sex and gender | Not applicable. |
| Reporting on race, ethnicity, or other socially relevant groupings | Not applicable. |
| Population characteristics | Not applicable. |
| Recruitment | Not applicable. |
| Ethics oversight | Not applicable. |

Note that full information on the approval of the study protocol must also be provided in the manuscript.

# Field-specific reporting

Please select the one below that is the best fit for your research. If you are not sure, read the appropriate sections before making your selection.

☒ Life sciences    ☐ Behavioural & social sciences    ☐ Ecological, evolutionary & environmental sciences

For a reference copy of the document with all sections, see [nature.com/documents/nr-reporting-summary-flat.pdf](http://nature.com/documents/nr-reporting-summary-flat.pdf)

# Life sciences study design

All studies must disclose on these points even when the disclosure is negative.

| | |
|---|---|
| Sample size | All individuals with RNA-sequencing and phased whole genome sequencing data from the Geuvadis consortium were included. All genes that were found to have at least one significant eQTL association in the Geuvadis analysis were included. |
| Data exclusions | No data were excluded from the analyses. |
| Replication | We replicated our findings using four different sequence-to-expression deep learning models. |
| Randomization | Samples were randomly divided into training/validation sets during cross validation for the PrediXcan-style elastic net model. |
| Blinding | No blinding was used in this observational study. |

# Reporting for specific materials, systems and methods

We require information from authors about some types of materials, experimental systems and methods used in many studies. Here, indicate whether each material, system or method listed is relevant to your study. If you are not sure if a list item applies to your research, read the appropriate section before selecting a response.

## Materials & experimental systems

| n/a | Involved in the study |
|---|---|
| ☒ | ☐ Antibodies |
| ☒ | ☐ Eukaryotic cell lines |
| ☒ | ☐ Palaeontology and archaeology |
| ☒ | ☐ Animals and other organisms |
| ☒ | ☐ Clinical data |
| ☒ | ☐ Dual use research of concern |
| ☒ | ☐ Plants |

## Methods

| n/a | Involved in the study |
|---|---|
| ☒ | ☐ ChIP-seq |
| ☒ | ☐ Flow cytometry |
| ☒ | ☐ MRI-based neuroimaging |

