## [Peer Review File · Nature Genetics]

Peer Review Information

Manuscript Title: Personal transcriptome variation is poorly explained by current genomic deep learning models

Corresponding author name(s): Professor Nilah Ioannidis

Reviewer Comments & Decisions:

Decision Letter, initial version:

12th Jun 2023

Dear Nilah,

Your Brief Communication, "Personal transcriptome variation is poorly explained by current genomic deep learning models" has now been seen by 3 referees. You will see from their comments below that while they find your work of interest, some important points are raised. We are interested in the possibility of publishing your study in Nature Genetics, but would like to consider your response to these concerns in the form of a revised manuscript before we make a final decision on publication.

In brief, the three reviews are all positive and sound supportive of a publication at the journal. There are a few suggestions made for additional analysis from Reviewers #1 and #2; to our reading, these are either not massively onerous to do (e.g. #2's comments on analysing the "most reliable" eQTL variants), would be of illustrative interest (predicting ATAC signal), or are suggested rather than required (Referee #1's comments). Hence, while we think all of these additions would strengthen the conclusions of your study, we leave it up to you and your co-authors to decide how much to fulfil given the acknowledged timeliness of your work means it behooves us to publish sooner rather than later!

To guide the scope of the revisions, the editors discuss the referee reports in detail within the team, including with the chief editor, with a view to identifying key priorities that should be addressed in revision and sometimes overruling referee requests that are deemed beyond the scope of the current study. We hope that you will find the prioritized set of referee points to be useful when revising your study. Please do not hesitate to get in touch if you would like to discuss these issues further.

We therefore invite you to revise your manuscript taking into account all reviewer and editor comments. Please highlight all changes in the manuscript text file. At this stage we will need you to upload a copy of the manuscript in MS Word .docx or similar editable format.

We are committed to providing a fair and constructive peer-review process. Do not hesitate to contact

us if there are specific requests from the reviewers that you believe are technically impossible or unlikely to yield a meaningful outcome.

*2) If you have not done so already please begin to revise your manuscript so that it conforms to our Brief Communication format instructions, available [here](http://www.nature.com/ng/authors/article_types/index.html). Refer also to any guidelines provided in this letter.

[REDACTED]

We hope to receive your revised manuscript within four to eight weeks. If you cannot send it within this time, please let us know.

Sincerely,

Michael Fletcher, PhD
Senior Editor, Nature Genetics

ORCID: 0000-0003-1589-7087

Referee expertise:

Referee #1: genomics; machine learning.

Referee #2: genetics; computational biology; genomics.

Referee #3: human genetics; gene expression analysis.

Reviewers' Comments:

Reviewer #1:

Remarks to the Author:

This manuscript presents an analysis that evaluates the ability of current genomic deep learning models to explain expression variation across individuals. The authors assess the performance of four deep learning (DL) models on the expression data from the Geuvadis consortium and discovered that current models are unable to predict the direction of variant effects. They found that the model's predictions have high correlations with expression values for each individual when the analysis looks across genes. However, when evaluating the correlations for a given gene across individuals, the results show that current DL models are unable to predict the effect of SNVs on gene expression. They also suggest that this underperformance is more likely due to modeling noise instead of an inherit mechanism complexity for a certain subset of genes.

The analysis results are comprehensive (for a Brief Communications) and they strongly support the authors' conclusions. The work is timely as deep learning models like Enformer shows great promise. However, evaluating these models in their use case is important to show their strengths and as shown in this paper, some of their limitations. Hence, this work would be of high value to the broader scientific community, especially to those who would be end users of Enformer and ML developers to consider key limitations of their current practices. With that said, there are a few points that if addressed could make the arguments stronger. Since this is a Brief Communications, some of the suggestions below may fall outside of the scope. Nevertheless, it may still be worth mentioning to some extent as other readers will likely have similar questions/concerns.

Major comments:

1. PrediXcan was trained on the variant data, while the DL models were evaluated on a zero-shot generalization of single-nucleotide variant effects. It would be interesting to explore whether directly training on the same dataset splits as PrediXcan can resolve the directionality issue. One strategy would be to use pretrained representations and train a linear model like PrediXcan on the average

representations, using the same data splits.

2. It would be interesting to show if the average absolute correlation from DL models reaches similar performance as PrediXcan. This can measure how well the magnitude of the effect sizes is associated with eQTL effect sizes.

3. Here, gene expression is measured via RNA-seq while the original models, i.e. enformer and basenji, were trained on CAGE-seq data. The correlation of RNA-seq and CAGE is sometimes high (Yu et al. NAR 2015) and sometimes not (Kawaji et al. Genome Res 2014). For the eQTLs in this study, how well do CAGE tracks correlate with RNA-seq (both should be within fantom5)? I'm wondering if the observed effect is due to differences in CAGE and RNA-seq or because of personalized genomes.

4. It might be beneficial to stratify eQTL results based on mRNA expression levels. This might help to address questions about when the directions are captured well (and when they are not)?

5. Enformer does well at predicting single-nucleotide mutations within CREs as measured via MPRAS from the CAGI5 challenge. How does one reconcile these results with the ones observed in this study?

6. There isn't much discussion on what could be root cause of not capturing variant effect directionality. Also, there isn't much discussed on how to build upon this knowledge and improve DL models.

7. It could be informative to probe what features are being learned by PrediXcan and not in DL models?

Minor comments:

1. Although a Github link is provided, the code repository is not available at the time of review.

Reviewer #2:

Remarks to the Author:

In this brief communication, Huang et al show that the state-of-the-art tools for predicting genomic tracks from sequence do a rather poor job in predicting the effects of sequence variation on gene expression - including predicting the direction of expression effect.

I think this is a very important observation that deserves publication, and I believe that the analysis was done well.

My main comment relates to the fact that the authors seem to consider eQTL data as a gold standard, but this doesn't necessarily have to be the case everywhere. In particular, the fact that deep learning tools do a somewhat better job for stronger eQTLs (even though they still do get the direction of effect wrong sometimes) suggests that in some cases at least, errors in eQTL detection may be the culprit. I don't think this affects the paper's overall conclusions - because it is clear from the analyses presented that eQTL data may not be entirely to blame - but I would perhaps consider further QC'ing subsetting eQTL data for the most "reliable" variants - such as those that are replicated by GTEx, for example - and repeating the analysis.

As a potential additional analysis, the authors could consider how well these tools predict ATAC-seq signals at these genes' promoters - which are good enough proxies of expression activity. Could it be that for whatever reason ATAC-seq data from a multi-individual dataset (such as one from Tehranchi et al., eLife 2019) are more amenable to prediction?

Reviewer #3:

Remarks to the Author:

In this paper, the authors use whole genome sequencing and gene expression data from the GEUVADIS cohort to assess the performance of four pre-trained deep learning models (Enformer, Basenji2, ExPecto, Xpresso) in predicting gene expression levels from personal whole genomes. This is a clever strategy that complements existing benchmarks that have focussed on fine mapped causal variants and measurements from reporter assays. One of the main findings of the paper is that, for most genes, the gene expression level predicted based on the personal genome sequence of the individual in the cis region of the gene correlates poorly with the measured gene expression level. Furthermore, different models often give predictions with opposite effect size direction. I think the result is interesting and the analyses in the paper have been conducted carefully and I do not have any major concerns.

Author Rebuttal to Initial comments

We are grateful to the reviewers for their time and effort reviewing this manuscript, and we appreciate their positive comments and constructive feedback. We have included more detailed responses to each point below.

Reviewer #1:

Remarks to the Author:

This manuscript presents an analysis that evaluates the ability of current genomic deep learning models to explain expression variation across individuals. The authors assess the performance of four deep learning (DL) models on the expression data from the Geuvadis consortium and discovered that current models are unable to predict the direction of variant effects. They found that the model's predictions have high correlations with expression values for each individual when the analysis looks across genes. However, when evaluating the correlations for a given gene across individuals, the results show that current DL models are unable to predict the effect of SNVs on gene expression. They also suggest that this underperformance is more likely due to modeling noise instead of an inherit mechanism complexity for a certain subset of genes.

The analysis results are comprehensive (for a Brief Communications) and they strongly support the authors' conclusions. The work is timely as deep learning models like Enformer shows great promise. However, evaluating these models in their use case is important to show their strengths and as shown in this paper, some of their limitations. Hence, this work would be of high value to the broader scientific community, especially to those who would be end users of Enformer and ML developers to consider key limitations of their current practices. With that said, there are a few points that if addressed could make the arguments stronger.

Since this is a Brief Communications, some of the suggestions below may fall outside of the scope. Nevertheless, it may still be worth mentioning to some extent as other readers will likely have similar questions/concerns.

We thank the reviewer for the positive feedback and have mentioned many of the points below in our revised manuscript.

Major comments:

1. PrediXcan was trained on the variant data, while the DL models were evaluated on a zero-shot generalization of single-nucleotide variant effects. It would be interesting to explore whether directly training on the same dataset splits as PrediXcan can resolve the directionality issue. One strategy would be to use pretrained representations and train a linear model like PrediXcan on the average representations, using the same data splits.

We agree that this difference in training strategy is a key distinction between the sequence-based DL models, which attempt to learn causal features of a sequence that can generalize to unseen sequences, and PrediXcan, which fits effect sizes (and directions of effect) for only the set of variants seen in its training set. This approach makes PrediXcan useful for applications such as transcriptome-wide association studies, which rely on accurate estimates of common variant effect sizes, but not for generalizing to unseen variants, which is necessary to interpret rare variants in personal genomes. Since PrediXcan has different goals and use cases, we

include it in the manuscript not as a competitor method, but rather as a baseline for how much genetic contribution to expression should be possible to learn for each gene in the dataset. We have clarified this distinction in the revised manuscript. While we agree that applying a PrediXcan-style gene-by-gene training strategy on the embeddings of the DL models could be interesting to explore, we expect that the resulting models would learn corrected effects and directionalities only for variants seen in the training individuals, and would thus have similar limitations to PrediXcan in terms of generalizability. Such models may no longer have a general understanding of causal sequence features, having instead memorized particular variant effects. We do agree that comprehensively exploring strategies for incorporating more sequence diversity into training genomic DL models while preserving generalizability is an important avenue of future work in this field, but beyond the scope of the current manuscript.

2. It would be interesting to show if the average absolute correlation from DL models reaches similar performance as PrediXcan. This can measure how well the magnitude of the effect sizes is associated with eQTL effect sizes.

We believe the reviewer is referring to the comparisons between PrediXcan and the four DL models shown in Fig. S6g-j (previously Fig. S5g-j in the original manuscript). Considering the absolute value of the y-axis in each plot, these data show that the absolute correlations obtained by the DL models still tend to be lower than the correlations obtained by PrediXcan for the majority of genes. For example, we include a plot below with absolute value on the y-axis for the Enformer-PrediXcan comparison:

Comparison of absolute value of Enformer cross-individual Spearman correlation and PrediXcan cross-individual Spearman correlation for each gene with a kernel density estimate overlaid.

We have also added a new figure (Fig. S4 in the revised manuscript) showing the histograms of absolute correlations for each DL model, illustrating that the mean absolute correlation in each case is still lower than the mean correlation for PrediXcan in Fig. S3e.

3. Here, gene expression is measured via RNA-seq while the original models, i.e. enformer and basenji, were trained on CAGE-seq data. The correlation of RNA-seq and CAGE is sometimes high (Yu et al. NAR 2015) and sometimes not (Kawaji et al. Genome Res 2014). For the eQTLs in this study, how well do CAGE tracks correlate with RNA-seq (both should be within fantom5)? I'm wondering if the observed effect is due to differences in CAGE and RNA-seq or because of personalized genomes.

The reviewer is correct that there are differences between CAGE and RNA-seq measurements of gene expression; however, these differences should affect both the cross-gene and cross-individual correlations computed here for personal genomes, and are thus

unlikely to explain the poor cross-individual performance. In particular, the cross-gene correlations that we compute on personal genomes are comparable to previously reported reference genome performances for all models, while the cross-individual correlations are close to zero. In addition, only two of the four tested models (Enformer and Basenji2) were trained on CAGE data, while the other two tested models (ExPecto and Xpresso) were trained on RNA-seq data and show the same pattern of results.

4. It might be beneficial to stratify eQTL results based on mRNA expression levels. This might help to address questions about when the directions are captured well (and when they are not)?

We believe this question is addressed by our analysis in Fig. 2c and Fig. S12 (previously Fig. S11), where we compare model performance to the mean expression level of each gene and find that the correlation between performance and expression level is close to zero for all models. Thus, it seems that the overall gene expression level is not a major factor in the ability of these models to predict accurate variant effects.

5. Enformer does well at predicting single-nucleotide mutations within CREs as measured via MPRA from the CAGI5 challenge. How does one reconcile these results with the ones observed in this study?

This is an interesting point related to different approaches for measuring variant effects when evaluating model predictions, which each have advantages and disadvantages. Previous evaluations have primarily focused on the ability of these models to predict effects of individual variants, as measured in eQTL or MPRA studies. However, it is difficult to identify the causal variants in eQTL studies, leading to inaccurate causal effect size estimates for many variants, and MPRA lacks the complex genomic and chromatin context of endogenous gene expression. To our knowledge, our manuscript and the co-submitted manuscript by Sasse, Ng, Spiro *et al* are the first to use personal genome sequences to evaluate performance, in which all variants surrounding each gene TSS are input together and evaluated on the corresponding expression measurement from that individual, avoiding the issue of causal variant identification. We speculate that the MPRA task mentioned by the reviewer, which primarily tested promoter variants, may be an easier task than correctly ordering the collective effects of many variants within a larger endogenous context around the TSS, which is needed for accurate personal gene expression prediction across individuals. Comparing and contrasting the results from multiple complementary methods of evaluating variant effect predictions will be an important component of future work.

6. There isn't much discussion on what could be root cause of not capturing variant effect directionality. Also, there isn't much discussed on how to build upon this knowledge and improve DL models.

We have added additional discussion to the manuscript, including some of the points mentioned above and in the response to reviewer 2 below.

7. It could be informative to probe what features are being learned by PrediXcan and not in DL models?

Since PrediXcan is a linear combination of variant dosages, its learned features are the estimated weights, or effect sizes, for these variants. However, these weights are not necessarily the true causal effects of the corresponding variants, due to linkage disequilibrium within the training set, and thus are not directly useful for evaluating the variant effect predictions from DL models. In addition, as mentioned above, PrediXcan uses a training strategy that learns weights only for the variants seen in its training individuals, and thus its features are not generalizable to new variants or sequences.

Minor comments:

1. Although a Github link is provided, the code repository is not available at the time of review.

We have made the github repository public.

Reviewer #2:

Remarks to the Author:

In this brief communication, Huang et al show that the state-of-the-art tools for predicting genomic tracks from sequence do a rather poor job in predicting the effects of sequence variation on gene expression - including predicting the direction of expression effect.

I think this is a very important observation that deserves publication, and I believe that the analysis was done well.

We thank the reviewer for the positive comments.

My main comment relates to the fact that the authors seem to consider eQTL data as a gold standard, but this doesn't necessarily have to be the case everywhere. In particular, the fact that deep learning tools do a somewhat better job for stronger eQTLs (even though they still

do get the direction of effect wrong sometimes) suggests that in some cases at least, errors in eQTL detection may be the culprit. I don't think this affects the paper's overall conclusions - because it is clear from the analyses presented that eQTL data may not be entirely to blame - but I would

perhaps consider further QC'ing subsetting eQTL data for the most "reliable" variants - such as those that are replicated by GTEx, for example - and repeating the analysis.

We agree with this important point about errors in eQTL detection, since the effect sizes estimated in eQTL studies are not biologically meaningful for variants that are non-causal, and thus non-causal eQTL variants should not be used to evaluate model variant effect predictions. Since it is difficult to determine the causal variants in eQTL studies, even with current fine-mapping approaches, evaluation sets based on eQTLs are likely to contain some incorrect variant effects. This issue highlights one of the advantages of our approach, in contrast to evaluations that rely on eQTL data as a gold standard. In particular, by using personal genome sequences to evaluate model performance, and thus including all variants from each individual within the input sequences to the models, our input sequences will contain any relevant causal variant(s) for each individual without needing to first identify those variants. Our approach thus avoids the issue of causal variant identification. We have added text to the manuscript to clarify this point.

As a potential additional analysis, the authors could consider how well these tools predict ATAC-seq signals at these genes' promoters - which are good enough proxies of expression activity. Could it be that for whatever reason ATAC-seq data from a multi-individual dataset (such as one from Tehranchi *et al.*, eLife 2019) are more amenable to prediction?

This is a good suggestion and is in line with our planned future work to characterize model performance and uncertainty when predicting cross-individual differences in additional molecular phenotypes, not just gene expression. We have done two preliminary analyses based on the reviewer's comment.

First, although the Tehranchi *et al.* ATAC-seq measurements are on pooled individuals and thus cannot be used to directly evaluate cross-individual performance on ground truth accessibility data, their analyses provide a set of chromatin accessibility QTL variants (caQTLs) that can be used to evaluate model predictions similarly to eQTL variants. For computational efficiency, we computed predictions for a random sample of 10,000 of the Tehranchi *et al.* caQTLs using Enformer, and evaluated Enformer's accuracy at predicting which of the two alleles has greater accessibility (Fig. S15 in the revised manuscript). We find moderate accuracy that increases somewhat when we subset to caQTLs detected in several

populations, which may be more likely to be true causal variants. These results are in line with an analysis in Zhou *et al* 2015, which used allelically imbalanced variants from DNase-seq data to evaluate DeepSEA's ability to predict the allele with more accessibility, and found similar accuracy. It is difficult to directly compare accuracies between caQTL and eQTL prediction tasks due to differences in the methods and thresholds used to call these QTLs, but these preliminary results suggest that current models still have room for improvement when predicting directions of effect on accessibility as well as expression. Further analysis is needed to more systematically understand the differences between chromatin accessibility and gene expression prediction.

Second, even without ground truth accessibility data we can compare model predictions of accessibility across individuals at each gene TSS to the same predictions from other deep learning models to evaluate their consistency. In a preliminary analysis, we compute correlations between the predictions of two models, Basenji2 and ExPecto, for accessibility and for expression at each gene TSS for genes whose most significant eQTL is located within 1kb of the TSS. We find that the correlation between models is similar for accessibility predictions and expression predictions:

For each gene, we plot the Spearman correlation between the Basenji2 predictions of DNase accessibility and ExPecto predictions of DNase accessibility within 700bp of the TSS across individuals (y-axis) and the Spearman correlation between Basenji2 and ExPecto predictions of gene expression across individuals (x-axis), colored by density. Mean correlations are shown along each axis.

This preliminary result suggests that variant effects on accessibility at the TSS are not significantly easier to predict than variant effects on expression; however, a more comprehensive evaluation of this question for accessibility and other phenotypes, predicted at the TSS and at the locations of putative causal variants, is planned for future work and is beyond the scope of the current manuscript. We have added a discussion of this point to the manuscript.

Reviewer #3:

Remarks to the Author:

In this paper, the authors use whole genome sequencing and gene expression data from the GEUVADIS cohort to assess the performance of four pre-trained deep learning models (Enformer, Basenji2, ExPecto, Xpresso) in predicting gene expression levels from personal whole genomes. This is a clever strategy that complements existing benchmarks that have focussed on fine mapped causal variants and measurements from reporter assays. One of the main findings of the paper is that, for most genes, the gene expression level predicted based on the personal genome sequence of the individual in the cis region of the gene correlates poorly with the measured gene expression level. Furthermore, different models often give predictions with opposite effect size direction. I think the result is interesting and the analyses in the paper have been conducted carefully and I do not have any major concerns.

We thank the reviewer for their comments and the positive feedback on this evaluation strategy in relation to existing benchmarks.

Decision Letter, first revision:

11th Jul 2023

Dear Nilah,

Thank you for submitting your revised manuscript "Personal transcriptome variation is poorly explained by current genomic deep learning models" (NG-BC62460R). It has now been seen by the original referees and their comments are below. The reviewers find that the paper has improved in revision, and therefore we'll be happy in principle to publish it in Nature Genetics, pending minor revisions to satisfy the referees' final requests and to comply with our editorial and formatting guidelines.

The current version of your manuscript is in a PDF format.

Please email us a copy of the file in an editable format (Microsoft Word or LaTeX)-- we can not proceed with PDFs at this stage.

Thank you again for your interest in Nature Genetics. Please do not hesitate to contact me if you have any questions.

Sincerely,

Michael Fletcher, PhD
Senior Editor, Nature Genetics

ORCID: 0000-0003-1589-7087

Reviewer #1 (Remarks to the Author):

The authors have addressed my concerns.

Reviewer #2 (Remarks to the Author):

I am happy with how the authors addressed my comments and recommend accepting the revised version.

Reviewer #3 (Remarks to the Author):

I have no further comments.

Reviewer #4 (Remarks to the Author):

The authors have addressed my concerns.

Final Decision Letter:

18th Oct 2023

Dear Nilah,

I am delighted to say that your manuscript "Personal transcriptome variation is poorly explained by current genomic deep learning models" has been accepted for publication in an upcoming issue of Nature Genetics.

Your paper will be published online after we receive your corrections and will appear in print in the next available issue. You can find out your date of online publication by contacting the Nature Press Office (press@nature.com) after sending your e-proof corrections. Now is the time to inform your Public Relations or Press Office about your paper, as they might be interested in promoting its publication. This will allow them time to prepare an accurate and satisfactory press release. Include your manuscript tracking number (NG-BC62460R1) and the name of the journal, which they will need when they contact our Press Office.

Please note that *Nature Genetics* is a Transformative Journal (TJ). Authors may publish their research with us through the traditional subscription access route or make their paper immediately open access through payment of an article-processing charge (APC). Authors will not be required to make a final decision about access to their article until it has been accepted. [Find out more about Transformative Journals](https://www.springernature.com/gp/open-research/transformative-journals)

Authors may need to take specific actions to achieve [compliance](https://www.springernature.com/gp/open-research/funding/policy-compliance-faqs) with funder and institutional open access mandates. If your research is supported by a funder that requires immediate open access (e.g. according to [Plan S principles](https://www.springernature.com/gp/open-research/plan-s-compliance))

then you should select the gold OA route, and we will direct you to the compliant route where possible. For authors selecting the subscription publication route, the journal's standard licensing terms will need to be accepted, including <https://www.nature.com/nature-portfolio/editorial-policies/self-archiving-and-license-to-publish>. Those licensing terms will supersede any other terms that the author or any third party may assert apply to any version of the manuscript.

If you have not already done so, we invite you to upload the step-by-step protocols used in this manuscript to the Protocols Exchange, part of our on-line web resource, natureprotocols.com. If you complete the upload by the time you receive your manuscript proofs, we can insert links in your article that lead directly to the protocol details. Your protocol will be made freely available upon publication of your paper. By participating in natureprotocols.com, you are enabling researchers to more readily reproduce or adapt the methodology you use. [Natureprotocols.com](https://natureprotocols.com) is fully searchable, providing your protocols and paper with increased utility and visibility. Please submit your protocol to <https://protocolexchange.researchsquare.com/>. After entering your [nature.com](https://www.nature.com) username and password you will need to enter your manuscript number (NG-BC62460R1). Further information can be found at <https://www.nature.com/nature-portfolio/editorial-policies/reporting-standards#protocols>

Sincerely,

Michael Fletcher, PhD
Senior Editor, Nature Genetics

ORCID: 0000-0003-1589-7087